# Influence of a Natural Plant Antioxidant on the Ageing Process of Ethylene-norbornene Copolymer (Topas)

**DOI:** 10.3390/ijms22084018

**Published:** 2021-04-13

**Authors:** Anna Masek, Angelika Plota

**Affiliations:** Faculty of Chemistry, Institute of Polymer and Dye Technology, Lodz University of Technology, 90-924 Lodz, Poland; angelika.plota@gmail.com

**Keywords:** antioxidants, hesperidin, aging, ethylene-norbornene copolymer, stabilization

## Abstract

In the field of polymer technology, a variety of mainly synthetic additives are used to stabilize the materials during processing. However, natural compounds of plant origin can be a green alternative to chemicals such as synthetic polyphenols. An analysis of the effect of hesperidin on the aging behavior of ethylene-norbornene copolymer was performed. The evaluation of changes in the tested samples was possible by applying the following tests: determination of the surface energy and OIT values, mechanical properties analysis, colour change measurements, FT-IR and TGA analyses. The obtained results proved that hesperidin can be effectively used as natural stabilizer for polymers. Furthermore, as a result of this compound addition to Topas-silica composites, their surface and physico-mechanical properties have been improved and the resistance to aging significantly increased. Additionally, hesperidin can act as a dye or colour indicator and only few scientific reports describe a possibility of using flavonoids to detect changes in products during their service life, e.g., in food packaging. In the available literature, there is no information about the potential use of hesperidin as a stabilizer for cycloolefin copolymers. Therefore, this approach may contribute not only to the current state of knowledge, but also presents an eco-friendly solution that can be a good alternative to synthetic stabilizers.

## 1. Introduction

The global production of plastic products on an industrial scale started to expand in the 1940s and 1950s. Analyzing data from the last twenty years, the global average annual production of plastics has increased by almost 80%, and in 2018 this amount was around 359 million tonnes [1,2]. This is due to the fact that polymeric materials have found a wide range of applications in numerous sectors [3,4,5,6,7]. It is estimated that in Europe, the main industries that use plastics are packaging (38%), construction (21%), automotive (7%), electronics (6%), and the remaining 28% is agriculture, household goods, as well as medicine and other sectors [2]. Moreover, products like foams, fibers, adhesives or coatings are made from polymers and nowadays they have also a huge potential in various areas. The reason for so many deployments on an industrial scale is their easy processability, good durability and low costs of the production process [8,9]. On the other hand, such a large production of polymeric products leads to serious management and environmental pollution problems. Mostly, these are caused by their short lifespan. It was stated that about 40% of all these materials are characterized by the lifespan shorter than one month [10].

During service life, polymeric composites are often exposed to factors such as sunlight, ultraviolet radiation, elevated temperature, moisture and many others, which contribute to the progress of their degradation processes. As a result, their chemical, physical, aesthetic and functional properties are deteriorated and at some point they can no longer be used [11,12,13,14]. For this reason, polymer additives which are responsible for improving the performance during processing (e.g., during extrusion), practicality, resistance to aging and, consequently, degradation of the polymeric products are an important aspect of their manufacture and use, A polymer matrix without any additives is very rarely encountered, typically only in some medical products, whereas in most cases, they are used as modifiers of properties in a controllable way [15,16]. Commonly used additives in polymeric materials include plasticizers, crosslinking agents, antioxidants, flame retardants, light and thermal stabilizers, and pigments [17,18,19,20,21,22].

Furthermore, more and more scientists are describing the use of natural additives in their studies that also describe a beneficial influence on human health and environment. In the accessible literature, various types of stabilizers used in the polymer industry can be found, but the most important group is undoubtedly natural antioxidants with anti-aging properties. Additionally, polymeric products’ service life is dependent on their effectiveness [23,24,25,26,27,28,29,30]. One of the best-known representatives of this group are flavonoids. They are organic compounds that occur in plants, and in the polymer industry they can act as dyes, antioxidants, and as substances protecting against attacks by fungi and insects [31]. Flavonoids’ efficiency hinges on their solubility in the polymer matrix, degree of dispersion, thermal stability, and radical scavenging mechanism [25]. In general, they can interact with free radicals directly or indirectly, reducing the reactivity of these compounds. Moreover, their action as stabilizers in polymers includes working as radical scavengers [R*] in order to protect against degradation processes during the service life of the product or decomposing hydroperoxides to create stable alcohols, thanks to which a color formation is reduced and processing stability is provided. One of the popular antioxidants is hesperidin that is a naturally found flavanone glycoside. It is present in citrus-based fruits and is characterized by anti-aging activities. In general, compounds like hesperidin from the flavonoids group have become a pro-ecological, green alternative to synthetic polyphenols, phosphate derivatives or hindered amine light stabilizers. Moreover, in the case of products, in which very restrictive standards have to be complied, natural antioxidants can be used in order to enhance their environmentally-friendly profile [27,32,33,34].

Ethylene-norbornene copolymer (Topas) belongs to the cyclic olefins copolymers (COCs) that were first produced in 1950, but they have gained great attention during the last decade because of their special properties, which make them desirable as high-tech engineering plastic products. It should be highlighted that ethylene-norbornene is the most interesting and universal among all of the COCs [35,36,37]. Topas copolymer is characterized by very high-purity, amorphous transparency, and low permeability to nitrogen, oxygen or moisture. As a result of those properties, ethylene-norbornene copolymer has found the application in medical devices, pharmaceutical and food packaging, electronics and optics [28,38,39,40]. Nevertheless, products made from this material, are characterized by poor resistance to thermal oxidation and photodegradation processes in outdoor conditions. As a result, they lose their mechanical and surface properties, which means that they cannot be applied for the production of long-duration materials that are used in outdoor applications [39].

The aim of this study was to obtain an ethylene-norbornene copolymer blend with a natural flavonoid—hesperidin—and a filler in the form of silica and analyze the influence of this additive on aging behavior in comparison with samples without this antioxidant. In the available literature, there is no information about the potential use of hesperidin from citrus-based fruits as stabilizer for cycloolefin copolymers. Therefore, this approach may contribute not only to the current state of knowledge, but also encourages the use of environmentally friendly additives that are able to improve physico-mechanical properties of the polymeric materials.

## 2. Results and Discussion

### 2.1. Surface Free Energy (SFE)

A significant property for polymers is their surface quality. For this reason, the surface free energy (SFE), which is the sum of the polar and dispersive components, was defined for all prepared samples before and after 100, 200, 300 and 400 h of weathering (Figure 1). The SFE was calculated by applying the Owens-Wendt-Rabel-Kealble (OWRK) approach that consists in determining the contact angles using polar and non-polar liquids. In this case, it was distilled water, ethylene glycol and 1,4-diiodomethane and the obtained values are presented in Table 1. It should be highlighted that the SFE of polymers sometimes informs about their susceptiveness to aging.

Comparing the calculated surface free energy values of the reference samples, a slight decrease for Topas-silica and Topas-silica-hesperidin blends was observed. Moreover, analyzing the polar components before aging, a little increase of this value was noted for the sample with hesperidin content. In the available literature, the influence of hesperidin on polylactide (PLA)-biopolymer has been studied [32]. In the authors’ opinion, hesperidin can enhance the amount of polar groups. This probably occurs because this flavonoid is labile and migrates to the oxidation initiation site, i.e., to the surface of the polymeric material, whereby it causes a change in surface properties due to the presence of polar groups. However, in the case of hesperidin addition to Topas samples, this effect is not clearly apparent. For Topas-silica reference sample, the polar constituent was lower (0.8 mJ/m^2^) in comparison to the Topas-silica-hesperidin composite (3.78 mJ/m^2^). This could mean that before weathering aging this material had better resistance to oxidation, which was also confirmed by analyzing the value of contact angles in the case of distilled water as the measuring liquid (96.1°). When a contact angle is above 90°, it signifies that the polymeric material has hydrophobic nature and is less susceptibility to degradation process [41]. Therefore, the polar component for the pure mixture of Topas and Topas-silica-hesperidin was higher, for which the contact angle values with water were lower than 90°, which can mean their more hydrophilic character. However, comparing the polar constituents of the surface free energy and contact angle values after weathering, the tendency is reversed. While for pure Topas and Topas-silica hesperidin composites after 400 h of aging, the polar components are equal to almost zero, for Topas-silica mixture an increase of this value can be observed. Additionally, analyzing the contact angles, for Topas-silica material this value is equal to 87.9° after 400 h of weathering aging, which may indicate a change in the surface properties and its higher susceptibility to oxidation process, but it should be emphasized that the observed differences in surface energy results not considerable and could correspond simply to experimental errors. Therefore, in order to assess the influence of hesperidin content on aging behaviour of Topas-silica samples, other tests were also performed.

### 2.2. Fourier-Transform Infrared Spectroscopy (FT-IR) Analysis

The weathering progress was also examined by using Fourier-transform infrared spectroscopy that allows one to simultaneously analyze the functional groups present in materials and any structural changes generated by the exposure to higher temperature, radiation intensity, and humidity. Significant changes were observed for carbonyl groups (C=O) within the 1600–800 cm^−1^ range of the absorbance spectra of pure Topas (A), Topas-silica (B), Topas-silica-hesperidin (C) blends in Figure 2.

During FT-IR analysis, characteristic peaks belonging to the ethylene-norbornene copolymer were noted at 2848 and 2915 cm^−1^ assigned to the stretching vibrations of CH_2_ groups, at 1711 cm^−1^ corresponding to carbonyl groups (C=O), at 1463 cm^−1^ representing CH_2_ groups (bending band), and at 719 cm^−1^, which is a rocking band of CH_2_. Additionally, stretching vibrations of silica (SiO_2_) groups at 1093 cm^−1^ (Si-O-Si) and 806 cm^−1^ (Si-O) and one weak bending band at 471 cm^−1^ (Si-O) were observed. Unfortunately, the carbonyl (C=O) band of hesperidin did not appear at 1645 cm^−1^, which is a characteristic peak for this substance. This could be caused by dispersion of the hesperidin in the polymer matrix, which may be not at the molecular level. Moreover, it can be supposed that as a result of this phenomenon, very minor differences in surface free energy results were also observed.

However, the most significant change in peak intensity at 1711 cm^−1^ (C=O group) due to the weathering was noticed. Comparing the prepared Topas-based composites, the highest difference in this peak absorbance was found for Topas-silica mixture. Based on Equation (5), carbonyl index (CI) values for each material after aging were calculated (Figure 3). It can be said that the CI for pure Topas and Topas-silica-hesperidin samples is similar whereas in the case of Topas-silica composite, these values increase significantly with the duration of aging. A difference in carbonyl index between Topas-silica with and without hesperidin is equal to 150%. Therefore, based on this analysis, it can be assumed that the addition of hesperidin in the form of crystallites may have also contributed to the retardation of the aging process.

### 2.3. Change of Colour Measurement

The next step in this study was an evaluation of colour difference measurement for Topas-based composites performed by UV-Vis spectrophotometry. Colour changes are often the first visible sign of polymeric materials’ degradation and especially in optical appliances in which a transparency of the material is required, this is not an acceptable outcome. In some cases, however, a colour change can be recognized as an asset. Firstly, colourful additives applied in polymer products, can act as natural dyes because the addition of flavonoids to polymers can match the colour changes of the final polymeric products. Rangel de Sousa et al. (2016) [42] investigated the anti-UV aging action of beetroot and annatto extracts in an ultra-high molecular weight (UHMWPE) matrix. According to the authors, both the beetroot and annatto seed extracts can act efficiently as pigments for the same polymeric matrix. Moreover, more and more scientific papers describe a possibility of using natural antioxidants as successful stabilizers in active food packaging, in which they are able to slow down or prevent the oxidation processes of food components, such as lipids and proteins [43,44,45,46]. Masek et al. [28] applied pure quercetin as an eco-friendly color indicator in a Topas copolymer and their research showed that this compound changed the color of the materials under the influence of various factors, such as UV radiation, temperature, and humidity. Therefore, it can be used as a colour indicator of aging time of the polymeric products. This paper is focused on hesperidin as a potential dye and colour indicator in polymers. Table 2 shows the colour parameters (L*, a*, b*) of the Topas-based vulcanizates before aging that represent the differences between points plotted in the CIE-Lab space, which are related to the visual colour changes. Figure 4 presents the obtained results of colour difference (A), chroma (B), whiteness index (C), and hue angle (D) values for tested samples after 100, 200, 300 and 400 h of weathering aging including daytime and nighttime cycles.

In general, analyzing the colour parameters determined in the CIE-Lab colour space, the maximum value for L* is 100, which means a point-device reflecting diffuser whereas the minimum value for L* is zero, which indicates black. However, the a* and b* coordinates have no determinate numerical limitations. The positive value of a* means red colour and negative–green. The positive b* coordinate value indicates yellow colour and negative–blue. According to the Table 2, Topas sample containing silica and hesperidin is darker in comparison to the reference sample, as indicated by the lower value of parameter L*. Furthermore, this flavonoid shifted the colour of the Topas vulcanizate towards red, as indicated by the decrease in the value of a* parameter by 3.12, and towards yellow, as indicated by the increase in b* parameter by 13.45. Moreover, the colour change of Topas sample containing silica and hesperidin became distinctly apparent under the influence of UV radiation, elevated temperature and humidity (Figure 4).

The colour difference results showed that both pure Topas and the mixture of Topas with silica had a slight colouration change. The greatest impact of aging was observed for the Topas-silica-hesperidin sample, for which the colour difference after 200 and 300 h of weathering was close to 20. The same tendency was noted for the chroma (C_ab_) values. In the case of blends with the hesperidin content, there was a significant increase in their degree of saturation. Additionally, analyzing the whiteness index (W_i_) results for the same samples, it can be seen that they darkened. What interesting, the major changes for Topas-silica-hesperidin composites in optical properties were observed after 200 and 300 h of aging. It can be assumed that after 300 h of aging these samples underwent major degradation and more significant changes took place inside their structure, not on their surface. Moreover, all hue angles (h_ab_) results were close to 90°, which indicates that they are similar in colour to yellow.

The optical changes of all Topas samples are presented in Figure 5. The visual assessment confirms the results obtained above for the color change measurements of Topas composites, for which the greatest weathering effect was observed for the sample with a hesperidin addition after 200 and 300 h of aging. In general, when the colour difference value is below 2, it cannot be seen by human eyesight. Therefore, changes for pure Topas materials are practically imperceptible.

Flavonoids (chalcones, anthocyanins, flavones, aurones and flavanols) are chemical compounds that may differ from each other in the location or number of hydroxyl groups, the presence of dimeric structures and the degree of C ring oxidation. What is important, they are able to absorb the visible light in the range of 280–315 nm. However, the bioavailability of some flavonoids is limited because of their low solubility in water. For instance, Park [47] stated that α- and β-naphthoflavone (as examples of synthetic flavonoids) exhibit infinitesimal aqueous solubility due to their very low hydroxylation. On the other hand, these natural substances are characterized by good protection against UV radiation, antioxidant properties, good antibacterial activity and they are responsible for the colouration of fruits and flowers. The color of flavonoids depends on their chemical structure, especially the presence of hydroxyl and methoxyl groups. For example, compounds withnumerous OH groups may display a blue colouration, while OCH_3_ groups give these compounds a red colour, and as a result, when added to polymeric materials, they can significantly change the pigmentation of the final product. It was also noted that the characteristic feature of these substances is the colour change with environmental factors [27,48,49,50]. The flavonoid hesperidin used in this study belongs to the flavanones group which are derivatives of flavones that usually have a yellowish or cream tone, and absorb ultraviolet radiation, hence they are able to protect the polymer matrix. Therefore, colour change of composites with hesperidin content was observed, as a result of its sensitivity to UV radiation.

Furthermore, in the available literature, it was reported that the oxidation processes of polymers may change the molecular structure of antioxidants and thus also changes its color [28]. Therefore, polyphenolic compounds have not only the ability to absorb radiation, thanks to which they protect the polymer matrix, but also when used in polymeric products they can act as colorimetric indicators and detect changes caused by various factors during their service life.

### 2.4. Mechanical Properties

Furthermore, the analysis of mechanical properties of Topas-based mixtures before and after weathering aging was performed. All obtained results are presented in Table 3. Comparing the reference samples, the greatest values of tensile strength (σ), maximum stress (T_Fmax_), elongation at break (ε), and strain at maximum tensile strength (E_Fmax_) received for Topas-silica composites, which can be due to the fact that SiO_2_ is an active filler improving the physical and mechanical properties of polymeric products. In the case of the stresses values at 100, 200, and 300% of elongation, very similar results obtained for the composites with silica and silica with hesperidin content, where for pure Topas these stresses were lower. Moreover, it should be highlighted that for some samples, tensile strength (σ) and elongation at break (ε) cannot be assessed because they reached the limit of the extension range of the machine.

Despite the good mechanical properties of the reference sample with silica addition, the most considerable changes in mechanical properties were evidenced for these composites by a significant decrease in the tensile strength and elongation at break values after 400 h of aging, which was almost 75% in comparison to the reference mixture. This phenomenon could be due to the fact that the silica imparted stiffness and brittleness to the prepared composites. However, Topas-silica-hesperidin blends were characterized by much higher values of tensile strength and elongation at break. After 300 h of weathering the tensile strength decreased slightly (38.7 MPa), only after 400 h the greater influence of aging on the obtained value can be noted (33.2 MPa). Martinez de Arenaza et al. [51] observed that the incorporation of BaSO_4_ particles to polylactide (PLA) matrix is a good strategy to improve the toughness of PLA composites. Their research showed that the PLA filled with only 1 wt.% of BaSO_4_ was characterized by a dramatic increase in elongation at break value (~2844% higher than for the reference sample) and also in tensile strength (~14% higher than for pure PLA sample), what resulted in enhancing the toughness of the material. In authors opinion, this phenomenon was related to the debonding and crack deflection between the PLA matrix and the inorganic BaSO_4_ particles and it was responsible for the flexibility behavior of the tested samples. Analyzing the obtained values of tensile strength and elongation at break for all reference samples in the following study, this effect was not clearly apparent because the results for Topas-silica-hesperidin vulcanizate was only slightly higher than for pure Topas. Therefore, comparing the mechanical properties of Topas-silica composites with and without hesperidin content before and after aging, it can be assumed that this flavonoid, even in the form of crystallites, stunts the aging and stabilizes the polymer matrix, which can be seen by a slight change in mechanical features. The same tendency was described by Masek et al. [52], who investigated the effect of morin hydrate in the form of crystallites on the aging behaviour of Topas polymer. In authors opinion, this natural antioxidant from the flavonoid group applied for elastomeric materials can act as the anti-aging substance and protects the polymer matrix against the synergistic effects of climatic conditions.

In addition, for Topas-based composites, for which the values of tensile strength and elongation at break were determined, based on Equation (10), the aging coefficient (K) was calculated after 300 and 400 h of weathering (Figure 6). This coefficient informs about the degree of polymer degradation. When it is close to 1, a sample is said to have good resistance to aging. On the other hand, when this value is closer to 0, it means that it is more susceptible to degradation. Pure Topas and Topas-silica-hesperidin mixtures showed a good resistance to elevated temperature, radiation, and humidity in comparison to Topas with only silica addition, for which the aging coefficient was equal to 0.11 after 400 h of weathering. The sample with hesperidin content exhibited very high endurance until the aging time was 300 h (K = 0.98), after which its properties deteriorated (K = 0.72). What is interesting, the same tendency was observed for colour change measurements. It can be assumed that after 300 h of exposure to the synergistic effects of climatic conditions, hesperidin ability to absorb the UV radiation is no longer possible, and the polymer matrix starts to degrade.

### 2.5. Thermogravimetric Analysis (TGA)

Thermogravimetric analysis (TGA) was performed in order to assess a thermal stability of Topas-based samples (Figure 7 and Table 4). Table 4 presents the obtained values of T_2%_, T_5%_, T_10%_, T_20%_, T_50%_, T_70%_, and T_90%_ for pure Topas, Topas-silica and Topas-silica-hesperidin blends before weathering aging, which pertain to the mass change of 2%, 5%, 10%, 20%, 50%, 70%, and 90%, respectively.

The thermal decomposition of the all Topas mixtures occurred in one step, in which the initial temperature was around 432 °C and the maximum temperature was close to 490 °C. Moreover, the silica and silica with hesperidin contents did not have any influence on the thermal stability, which has been proven by the slight differences in the values in Table 4. The greatest change in temperatures for 90% of weight loss was observed for Topas-silica and Topas-silica-hesperidin composites (497.5 °C).

### 2.6. Oxidation Induction Time (OIT)

In addition, the oxidation induction time (OIT) values were determined as the time between melting and beginning of decomposition of the prepared mixtures in order to check a degree of thermal stabilization (Table 5). This test confirmed the TGA results because the natural additives did not improve this parameter. Moreover, in this case pure Topas was characterized by better thermal stability, for which the OIT was equal to 5.09 min while for Topas samples with silica and silica with hesperidin, this parameter was lower. On the other hand, comparing the Topas-silica and Topas-silica-hesperidin mixtures, the polyphenol leads to higher resistance to oxidation and the OIT value is 3.23 min when for ethylene-norbornene copolymer with silica is equal to 1.93 min. In the available literature, a comparison of the effects of different flavonoids (chrysin, quercetin, silibinin, naringin, and hesperidin) on oxidative degradation of polypropylene was performed [53]. This paper showed that hesperidin has no relevant oxidative retardant effect on the polypropylene compared to quercetin or silibinin, which improve considerably the OIT parameter. In the following research, the same tendency was observed, where the chosen flavonoid did not delay the onset of the thermal oxidation.

### 2.7. Scanning Electron Microscopy (SEM) Analysis

In order to evaluate the morphology of the prepared samples, scanning electron microscopy was carried out. This method allowed the examination of the dispersion as well as the size and amount of the phases and particles presented in Topas-based composites. Figure 8 shows the SEM images for hesperidin powder whereas Figure 9 presents the SEM images obtained for pure Topas, Topas-silica and Topas-silica-hesperidin mixtures. In both cases, three various magnifications were applied: 10,000×, 25,000×, 50,000×, respectively.

In Figure 9e,f a good dispersion of silica particles in Topas elastomer can be observed. This analysis proved that the flavonoid used in this study has a crystalline structure, which can be seen in Figure 9h,i. Moreover, comparing with Figure 8a,h, it is clearly apparent that hesperidin is in the form of crystallites, which may result in not very good dispersion of this antioxidant in the polymer matrix. Therefore, for the future research, it would be better to consider another processing method, in which the total dispersion of flavonoid can be achieved and it can be supposed that the effectiveness of the used additive with antioxidant properties would be improved.

## 3. Materials and Methods

### 3.1. Mixture Reagents

The polymer matrix in this study was ethylene–norbornene copolymer (TOPAS^®^ Elastomer E-140) from TOPAS Advanced Polymers^®^ (Raunheim, Germany), which is a thermoplastic elastomer and acts as a flexibilizer and impact modifier. It is characterized by the Vicat softening temperature—(64 °C) and the melting temperature—(84 °C). The content of bound norbornene was 40 wt.%. Moreover, this polymer is colorless, odorless, water-insoluble and has a good transparency.

Additionally, in two blends, the pyrogenic silica Aerosil 380 (Degussa, Germany) was applied as a filler. As a natural stabilizer, a hesperidin (hesperetin 7-rhamnoglucoside, ≥80%) from Sigma-Aldrich (Steinheim, Germany) was used as received. It belongs to flavonoids group that are characterized by good antioxidant properties. The structural formulas of ethylene-norbornene copolymer and hesperidin are shown in Figure 10.

### 3.2. Method of Topas Blends Preparation with Natural Additives

The polymer mixtures of ethylene–norbornene copolymer (Topas) were prepared in a laboratory micromixer (Brabender Lab-Station from Plasti-Corder with a Julabo cooling system, Duisburg, Germany) at 110 °C. The mixing time was 30 min, and the rotation speed was 60 rpm/min. After this time, the mixtures were vulcanized in two special steel molds with Teflon sheets in an electrically heated hydraulic press at a temperature of 160 °C under 125 bar of a pressure. The pressing time was 10 min. The compositions of all prepared mixtures are shown in Table 6.

### 3.3. Weathering Aging

Topas blends were subjected to the weathering aging test according to the PN-EN 4892-2 standard Plastics-Methods of exposure to laboratory light sources-Part 2: Xenon arc lamps”. This test was carried out by using a Weather-Ometer Ci 4000 chamber (Atlas Material Testing Technology LLC, Chicago, IL, USA), which is equipped with a xenon lamp. The operation mode of the machine consisted of two repeating segments by turns, called daytime and nighttime panels respectively, and each of them lasted 15 h. During daily cycle, the irradiation intensity was equal to 60.4 W/m^2^ (in radiation range of 300–400 nm), the temperature was 60 °C and the humidity was 80% with rainfall on. On the other hand, the nighttime part was characterized by no radiation, the temperature of 50 °C and humidity that was equal to 70% with rainfall off. Prepared samples were stowed in special stainless-steel frames and placed in the aging chamber. The test duration was 100, 200, 300, and 400 h, respectively, that was sufficient to see the changes that appeared in the tested materials.

### 3.4. Measurement Methods

#### 3.4.1. Surface Free Energy (SFE)

In order to define a surface free energy (SFE) of the prepared Topas blends before and after weathering aging, a goniometer OCA 15EC (DataPhysics Instruments GmbH, Filderstadt, Germany) was used. The SFE was determined on the basis of the Owens-Wendt-Rabel-Kealble (OWRK) method, which consists in measuring the contact angle of the tested material, using polar and non-polar liquids. In this method, the surface free energy of a solid is defined as the sum of the dispersive and polar components. The polar component is a sum of the hydrogen, acid-base and inductive forces, while a dispersive component is the magnitude of the intermolecular interactions (London forces). In order to calculate the SFE, the following Equations (1)–(4) were applied [54]:(1)$$E_{\mathrm{total}}=E_{\mathrm{polar}}+E_{\mathrm{dispersive}}$$
(2)$$\frac{\sigma_{L}\left(1+\cos\Theta \right)}{2\sqrt{\sigma_{L}^{D}}}=\sqrt{\sigma_{S}^{P}}\cdot \sqrt{\frac{\sigma_{L}^{P}}{\sigma_{L}^{D}}}+\sqrt{\sigma_{S}^{D}},\;Y=ax+b$$
(3)$$\mathrm{Whereas}:\;Y=\frac{\gamma_{L}\left(1+\cos\Theta \right)}{2\sqrt{\sigma_{L}^{D}}},\;X=\sqrt{\frac{\sigma_{L}^{P}}{\sigma_{L}^{D}}},\;a=\sqrt{\sigma_{S}^{P}},\;b=\sqrt{\sigma_{S}^{D}}$$
(4)$$\mathrm{Then}:\;E_{polar}=a^{2}=\sigma_{S}^{P}\;\mathrm{and}\;E_{dispersive}=b^{2}=\sigma_{S}^{D}$$
where $E_{\mathrm{total}}$ is a total surface free energy, which consists of polar and dispersive components respectively, $\sigma_{L}$ is the total surface tension of liquid, $\sigma_{L}^{P}$ and $\sigma_{L}^{D}$ are the polar and dispersive parts of the surface tension of liquid, $\sigma_{S}^{P}$ and $\sigma_{S}^{D}$ constitute the polar and dispersive components of the surface tension of solid, and Θ is a contact angle.

To measure contact angles of the Topas blends, three liquids with different polarities were used: distilled water, ethylene glycol, and 1,4-diiodomethane. For each sample, 15 contact angles of the three liquids were determined.

#### 3.4.2. Fourier-Transform Infrared Spectroscopy (FT-IR) Analysis

Fourier transform infrared spectroscopy (FT-IR) absorbance spectra were investigated within the 4000–400 cm^−1^ range by using a Nicolet 6700 FT-IR spectrometer (Thermo Scientific, Waltham, Massachusetts, USA) equipped with a diamond Smart Orbit ATR sampling tool. This research allowed to assess the structural changes that occurred as a result of weathering aging.

The FT-IR analysis enabled to monitor the oxidation process and observe changes in the carbonyl band (C=O). Based on the FT-IR spectrum, a carbonyl index (CI) for each sample after aging was calculated using Equation (5) [39]:(5)$$\mathrm{CI}=\frac{I_{C=O}}{I_{C-H}}$$
where I_C = O_ is the intensity of the peak that corresponds to the C=O groups and I_C-H_ is the intensity of the peak that represents the C-H groups.

The carbonyl index was applied to determine the absorption band of the carbonyl species that were formed during aging process within the 1600–1800 cm^−1^ range, by calculating a ratio of the carbonyl peak comparative to a reference peak [55].

#### 3.4.3. Change of Colour Measurement

According to the PN-EN ISO 105-J01 standard, the colour measurements of the tested materials were performed before and after aging using the UV-VIS CM-36001 spectrophotometer (Konica Minolta Sensing, Osaka, Japan). Then, the obtained results were interpreted in the CIE-Lab space, in which colour that is described by three parameters: L-lightness index from 0 (black) to 100 (white), a-coordinate that represents shades from green to red, b-coordinate corresponding to shades from blue to yellow. Based on the received results, the colour difference (ΔE), whiteness index (W_i_), chroma (C_ab_), and hue angle (h_ab_) values were calculated by using the Equations (6)–(9) [56]:(6)$$\Delta E=\sqrt{{\left(\Delta a\right)}^{2}+{\left(\Delta b\right)}^{2}+{\left(\Delta L\right)}^{2}}$$
(7)$$W_{i}=100-\sqrt{a^{2}+b^{2}+{\left(100-L\right)}^{2}}$$
(8)$$C_{\mathrm{ab}}=\sqrt{a^{2}+b^{2}}$$
(9)$$h_{\mathrm{ab}}=\left\{\begin{array}{c} \mathrm{arctg}\left(\frac{b}{a}\right)\;\mathrm{as}\;\;a>0\wedge b>0\\ 180{}^{\circ}+\mathrm{arctg}\left(\frac{b}{a}\right)\;\mathrm{as}\;\;\left(a<0\wedge b>0\right)\vee (a<0\wedge b<0\\ 360{}^{\circ}+\mathrm{arctg}\left(\frac{b}{a}\right)\;\;\mathrm{as}\;\;\;a>0\wedge b<0 \end{array}\right.$$

#### 3.4.4. Mechanical Properties

According to the PN-ISO 37:1998, the static mechanical properties characterization before and after aging of the Topas blends was performed by applying a Zwick Roell Z005 device (Zwick Roell, Ulm, Germany). The measured values were TS-tensile strength [MPa], Eb-elongation at break [%], T_Fmax_-maximum tensile stress [MPa], E_Fmax_-elongation at break for the maximum tensile strength [%]. Mechanical tests were done on “dumbbell” shape specimens (4 mm width and 1.5 mm thick) with three measurements for each sample. The test speed was 500 mm/min and the pre-force was about 0.1 N. Based on the obtained results of the tensile strength and elongation at break, the aging coefficient (A_f_) was determined with the following Equation (10) [57]:(10)$$A_{f}={\left(\mathrm{TS}\cdot E_{b}\right)}_{\mathrm{after}\;\mathrm{aging}}/{\left(\mathrm{TS}\cdot E_{b}\right)}_{\mathrm{before}\;\mathrm{aging}}$$
where TS is the tensile strength [MPa] and E_b_ is the elongation at break [%].

#### 3.4.5. Thermogravimetric Analysis (TGA)

Thermal degradation process that involves the mass loss of a material as a function of raising temperature was detected by thermogravimetric analysis (TGA). Mettler Toledo^®^ (Greifensee, Switzerland) equipment was applied in order to determine the initial and the maximum temperature of the thermal degradation process. Samples with a weight of approximately 6–8 mg were heated from 25 to 800 °C (with a heating rate-10 °C/min) under an argon atmosphere.

#### 3.4.6. Oxidative-Induction Time (OIT)

Oxidative-induction time (OIT) parameter was determined by using a Mettler Toledo DSC (Greifensee, Switzerland) device in order to assess a degree of thermal stabilization. The OIT was measured as the time between melting and beginning of decomposition of the tested samples in isothermal conditions. Topas-based reference samples with a weight of 6–8 mg were heated from the room temperature to the investigation temperature, 210 °C under an air atmosphere. The time of measurement was equal to 50 min. To ensure the accuracy of the OIT determination, two tests of each sample were performed.

#### 3.4.7. Scanning Electron Microscopy (SEM) Analysis

Based on the images obtained from the scanning electron microscope (SEM, Zeiss, ULTRA Plus, Oberchoken, Germany), the morphology of Topas-based composites and hesperidin powder was evaluated. Magnification was 10,000, 25,000 and 50,000×.

## 4. Conclusions

Nowadays, instead of synthetic additives, natural stabilizers that also exhibit beneficial influences on human health and the environment are being used increasingly. Among all of them, the most important group is undoubtedly natural antioxidants with anti-aging properties. Moreover, a lifetime of polymeric materials is dependent on their effectiveness.

In this study, hesperidin, a member of the natural polyphenols class was applied in a polymer matrix of ethylene-norbornene copolymer that was filled with silica. The obtained results evidenced that this flavonoid can be effectively used as a natural stabilizer for such polymeric products. What is important, as a result of hesperidin addition to Topas-silica composites, their surface and physico-mechanical properties have been preserved and the progress of aging has slowed down. Comparing the carbonyl index parameter for Topas-silica and Topas-silica-hesperidin composites, a difference in this value is equal to 150%, which means that this flavonoid turned out to be successful. The same trend was observed in the analysis of mechanical properties, where the calculated aging coefficient after 400 h of weathering for the Topas-silica mixture was 0.11, while for the sample with hesperidin content it was 0.72. On the other hand, this compound can act as a dye or colour indicator that is able to detect the changes of products caused by various factors during their service life, e.g., in food packaging materials.

Summarize, the compounds like hesperidin from flavonoids group can be a pro-ecological, green alternative to synthetic polyphenols, phosphate derivatives or hindered amine light stabilizers. Moreover, in the case of products, in which very restrictive standards have to be complied, natural antioxidants can be used in order to enhance their environmentally-friendly profile.

## Figures and Tables

**Figure 1 ijms-22-04018-f001:**
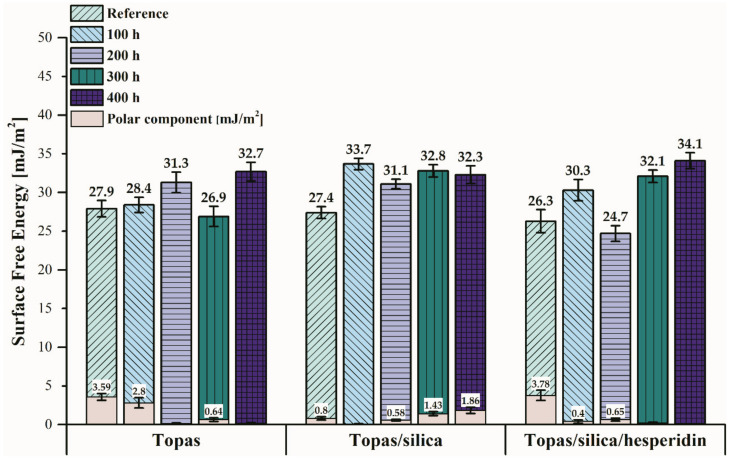
Changes in surface free energy (SFE) of Topas blends include the results before and after 100, 200, 300 and 400 h of aging, respectively.

**Figure 2 ijms-22-04018-f002:**
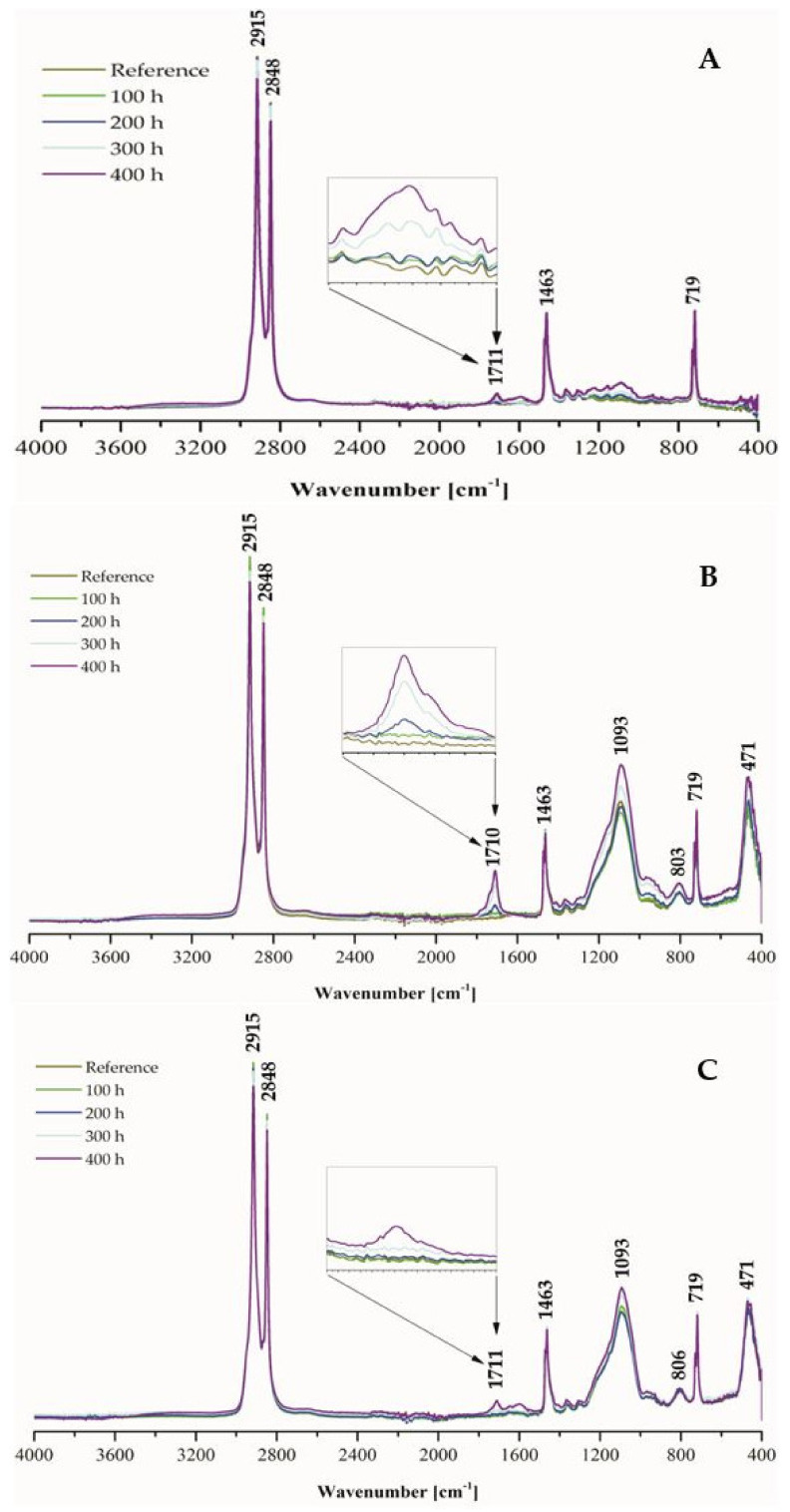
Fourier—transform infrared spectra of: (**A**) pure Topas, (**B**) Topas with silica, (**C**) Topas with silica and hesperidin, before and after 100, 200, 300 and 400 h of weathering.

**Figure 3 ijms-22-04018-f003:**
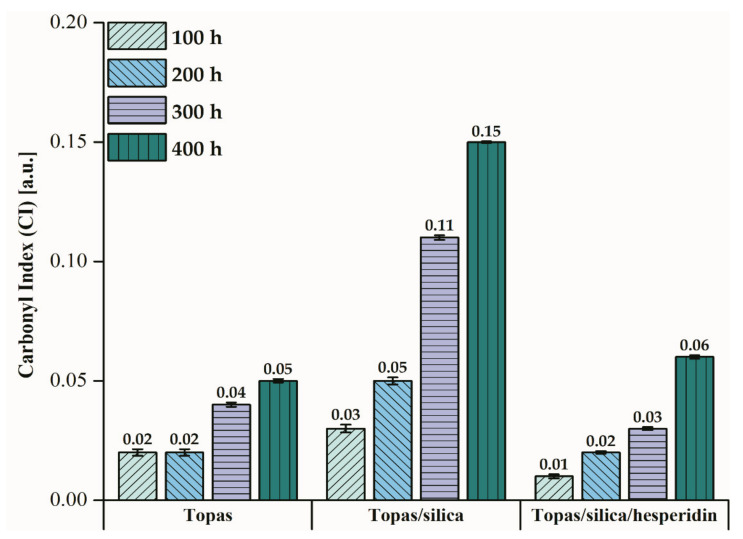
Carbonyl index (CI) values obtained for Topas-based mixtures after weathering, where C=O groups were observed within the 1600–1800 cm^−1^ range and CH groups at 2915 cm^−1^, respectively.

**Figure 4 ijms-22-04018-f004:**
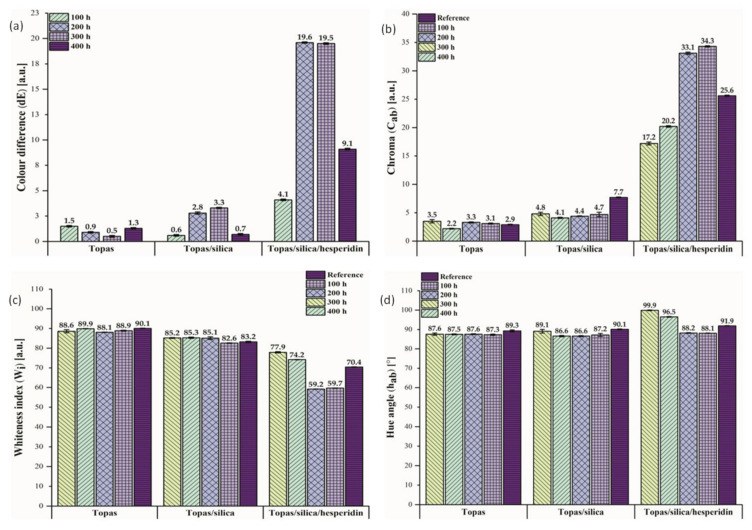
The influence of weathering on optical properties of Topas blends: (**a**) colour difference (dE), (**b**) chroma (C_ab_), (**c**) whiteness index (W_i_), (**d**) hue angle (h_ab_).

**Figure 5 ijms-22-04018-f005:**
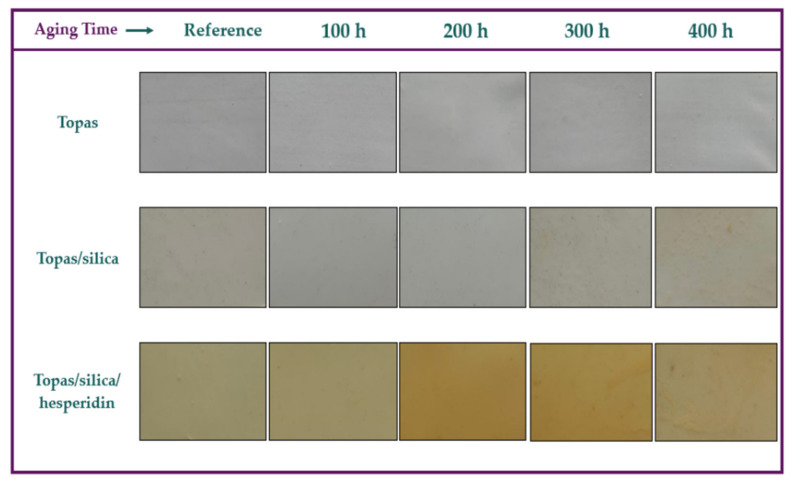
Visual changes in colour of prepared Topas samples induced by weathering aging.

**Figure 6 ijms-22-04018-f006:**
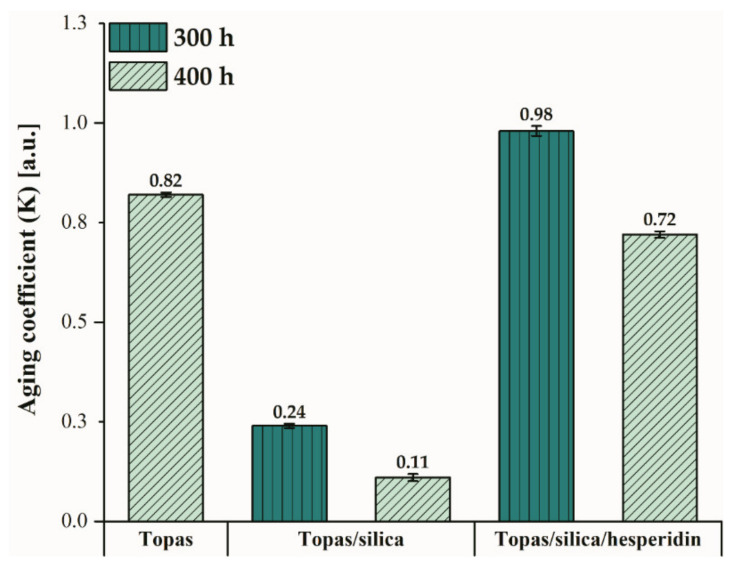
Aging coefficient values for pure Topas after 400 h of weathering and for Topas blends with silica and silica with hesperidin after 300 and 400 h of aging, respectively.

**Figure 7 ijms-22-04018-f007:**
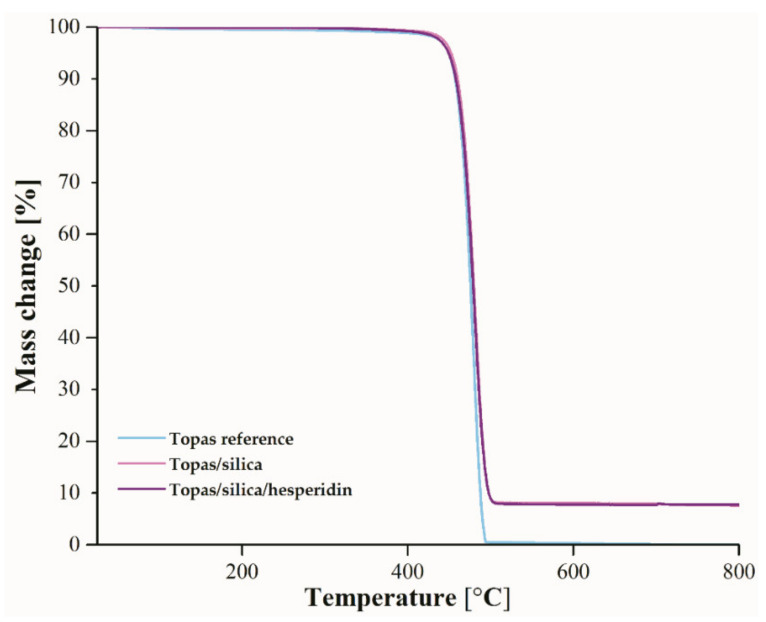
TGA curves of pure Topas and filled with silica and silica with hesperidin before aging that were heated from 25 to 800 °C.

**Figure 8 ijms-22-04018-f008:**
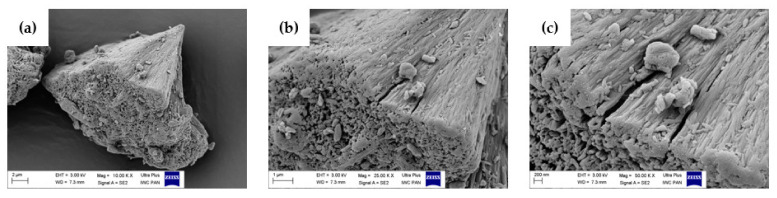
Morphology of hesperidin powder obtained by the scanning electron microscopy (SEM) with (**a**) 10,000×, (**b**) 25,000×, (**c**) 50,000× magnification, respectively.

**Figure 9 ijms-22-04018-f009:**
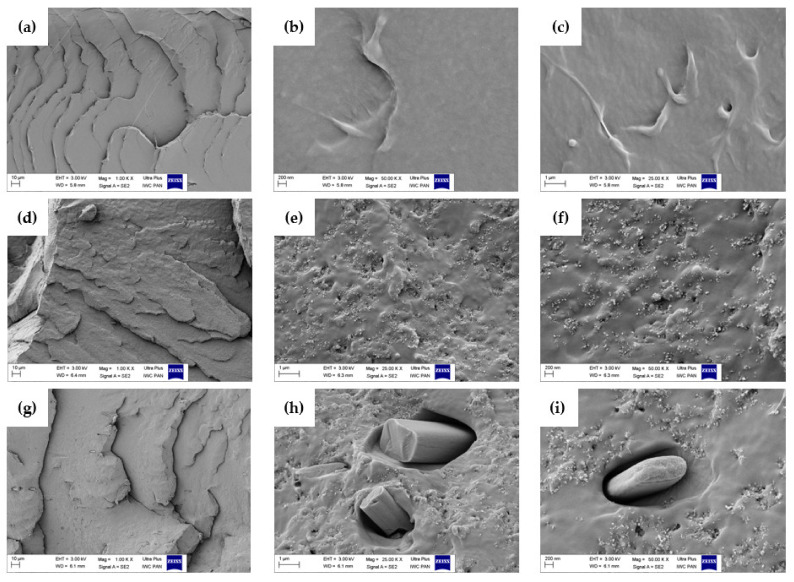
Morphology of (**a**–**c**) pure Topas, (**d**–**f**) Topas-silica, (**g**–**i**) Topas-silica-hesperidin composites obtained by the scanning electron microscopy (SEM) with 1000×, 25,000× and 50,000× magnification.

**Figure 10 ijms-22-04018-f010:**
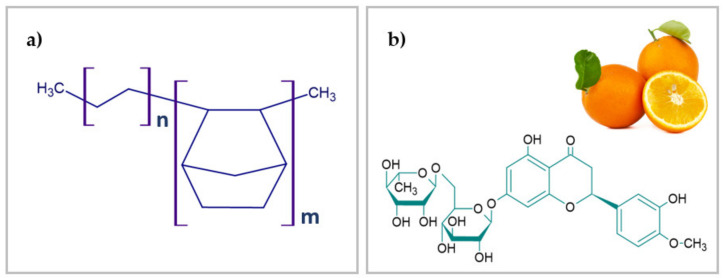
Structural formulas of: (**a**) polymer matrix—ethylene-norbornene copolymer (Topas), (**b**) natural flavonoid—hesperidin with antioxidant properties.

**Table 1 ijms-22-04018-t001:** Contact angle values determined by using distilled water, ethylene glycol, and 1,4-diiodomethane for Topas mixtures before and after weathering aging.

Liquid	Contact Angle after Weathering Aging [°]				
	Reference	100 h	200 h	300 h	400 h
**Topas**					
**Water**	87.6	90.6	98.2	97.8	97.6
**Diiodomethane**	61.6	59.9	55.4	62.5	53.8
**Ethylene glycol**	63.3	65.4	74.9	75.0	69.5
**Topas/Silica**					
**Water**	96.1	103.9	94.9	88.7	87.9
**Diiodomethane**	61.5	54.7	55.3	51.5	52.4
**Ethylene glycol**	73.6	78.7	69.7	65.7	64.1
**Topas/Silica/Hesperidin**					
**Water**	87.2	97.9	99.0	98.1	98.9
**Diiodomethane**	63.9	57.2	66.0	54.5	51.9
**Ethylene glycol**	67.2	71.9	78.1	71.0	73.4

**Table 2 ijms-22-04018-t002:** Influence of silica and hesperidin on the colour parameters (L*, a*, b*) of Topas vulcanizates before aging.

Sample	L* śr. [–]	a* śr. [–]	b* śr. [–]	ΔL [–]	Δa [–]	Δb [–]
Topas	89.13	0.15	3.47	0.74	0.05	0.29
Topas-silica	86.02	0.08	4.79	0.14	0.08	0.29
Topas-silica-hesperidin	86.21	−2.97	16.92	0.26	0.02	0.25

**Table 3 ijms-22-04018-t003:** Mechanical properties of samples based on ethylene-norbornene copolymer before and after weathering, where SE_100,200,300_ are the stresses at 100, 200, and 300% of elongation, respectively, T_Fmax_ is the maximum stress transferred by the sample [MPa], E_Fmax_ is the elongation at break for the maximum tensile stress [%], σ is a tensile strength [MPa], and ε is a total elongation at break [%].

Sample	Aging Time	SE_100_	SE_200_	SE_300_	T_Fmax_	E_Fmax_	σ	ε
	[h]	[MPa]	[MPa]	[MPa]	[MPa]	[%]	[MPa]	[%]
Topas	Reference	6.67	7.86	10.04	33.9	976	32.5	959.7
	100	7.13	8.48	10.60	34.9	993	-	-
	200	6.86	8.26	10.30	34.4	993	-	-
	300	6.84	8.02	10.00	34.8	994	-	-
	400	6.67	7.82	9.72	29.7	946	28.5	898.3
Topas/silica	Reference	7.42	8.89	11.40	40.8	989	39.4	980.2
	100	7.79	9.28	11.50	40.8	994	-	-
	200	8.30	9.70	11.70	25.8	728	25.3	729.2
	300	8.32	9.26	10.80	16.9	555	16.7	557.4
	400	7.98	8.47	9.47	10.3	369	9.8	374.4
Topas/silica/hesperidin	Reference	7.57	9.09	11.60	39.6	970	39.4	958.3
	100	7.75	9.26	11.70	41.5	993	-	-
	200	8.04	9.54	11.80	38.4	993	-	-
	300	8.31	9.96	12.30	38.7	952	38.7	951.9
	400	8.42	10.10	12.60	33.7	820	33.2	820.5

**Table 4 ijms-22-04018-t004:** Temperatures of the mass change of tested samples, where T_x%_ is a temperature at which the mass change is x% (2, 5, 10, 20, 50, 70, 90 %, respectively).

Mixture	Temperatures of Mass Change [°C]						
	T_2%_	T_5%_	T_10%_	T_20%_	T_50%_	T_70%_	T_90%_
**Topas**	432.5	449.2	456.7	464.2	475.8	480.8	487.5
**Topas/silica**	439.2	451.7	460.0	467.5	479.2	485.8	497.5
**Topas/silica/hesperidin**	434.2	449.2	457.5	465.8	478.3	485.8	497.5

**Table 5 ijms-22-04018-t005:** Oxidation induction time values of Topas-based samples before aging and their energy of oxidation.

Mixture	OIT Value [min]	Energy of Oxidation [J/g]
**Topas**	5.09	170
**Topas/silica**	1.93	247
**Topas/silica/hesperidin**	3.23	256

**Table 6 ijms-22-04018-t006:** The composition of Topas-based blends with natural antioxidant-hesperidin and a filler in the form of silica.

Mixture	Weight Composition [phr]		
	Topas	Silica A380	Hesperidin
**1**	100	-	-
**2**	100	10	-
**3**	100	10	1

## Data Availability

Not applicable.

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
