# Peer review of "Influence of a Natural Plant Antioxidant on the Ageing Process of Ethylene-norbornene Copolymer (Topas)"

_ijms, 2021, doi:10.3390/ijms22084018_

Round 1
Reviewer 1 Report
The paper "Influence of natural plant antioxidant on ageing process of Ethylene-norbornene Copolymer (Topas)" by Masek and Plota, investigated different properties of Topas copolymer, and of the same copolymer loaded with silica and with silica/hesperidin.
This article is a well-written work with a fairly large set of methods capable of confirming the authors' opinion. The data provided by the authors is believable. Many positions do not represent significant novelty. However, this remark does not prevent the publication of the work "as a whole" in its current state.
Author Response
Institute of Polymer and Dye Technology
Technical University of Lodz
90-924 Lodz, ul Stefanowskiego 12/16, Poland
Tel.: +48 42 631 32 23, Fax: +48 42 636 25 43
April 08, 2021
International Journal of Molecular Sciences
Dear Professor,
We are resubmitting our revised paper entitled “Influence of natural plant antioxidant on ageing process of Ethylene-norbornene Copolymer (Topas)” by with a request to reconsider it for publication in "International Journal of Molecular Sciences”.
We have carefully considered the Reviewers' comments. The manuscript was revised exactly according to these comments. The list of responses to the reviewer’s comments and corrections made in the manuscript are attached.
In the manuscript, the changes made based on the Reviewers' comments are marked in red.
The manuscript has not been previously published, is not currently submitted for review to any other journal, and will not be submitted elsewhere before a decision is made by this journal.
For correspondence please use the following information:
corresponding author: Anna Masek
Institute of Polymer and Dye Technology
Technical University of Lodz
90-924 Lodz, ul Stefanowskiego 12/16, Poland
Tel.: +48 42 631 32 93
Fax: +48 42 636 25 43
e-mail: anna.masek@p.lodz.pl
Yours sincerely,
PhD, Dsc Anna Masek
Answers to Reviewer #1 comments
Reviewer #1: The paper "Influence of natural plant antioxidant on ageing process of Ethylene-norbornene Copolymer (Topas)" by Masek and Plota, investigated different properties of Topas copolymer, and of the same copolymer loaded with silica and with silica/hesperidin.
This article is a well-written work with a fairly large set of methods capable of confirming the authors' opinion. The data provided by the authors is believable. Many positions do not represent significant novelty. However, this remark does not prevent the publication of the work "as a whole" in its current state.
Response: In order to present the results even better, we have also included a SEM analysis for both the composites and the antioxidant powder.
Reviewer 2 Report
In my opinion, the results do not support the conclusions, see attached pdf

Author Response
Institute of Polymer and Dye Technology
Technical University of Lodz
90-924 Lodz, ul Stefanowskiego 12/16, Poland
Tel.: +48 42 631 32 23, Fax: +48 42 636 25 43
April 08, 2021
International Journal of Molecular Sciences
Dear Professor,
We are resubmitting our revised paper entitled “Influence of natural plant antioxidant on ageing process of Ethylene-norbornene Copolymer (Topas)” by with a request to reconsider it for publication in "International Journal of Molecular Sciences”.
We have carefully considered the Reviewers' comments. The manuscript was revised exactly according to these comments. The list of responses to the reviewer’s comments and corrections made in the manuscript are attached.
In the manuscript, the changes made based on the Reviewers' comments are marked in red.
The manuscript has not been previously published, is not currently submitted for review to any other journal, and will not be submitted elsewhere before a decision is made by this journal.
For correspondence please use the following information:
corresponding author: Anna Masek
Institute of Polymer and Dye Technology
Technical University of Lodz
90-924 Lodz, ul Stefanowskiego 12/16, Poland
Tel.: +48 42 631 32 93
Fax: +48 42 636 25 43
e-mail: anna.masek@p.lodz.pl
Yours sincerely,
PhD, Dsc Anna Masek
Answers to Reviewer #2 comments
Reviewer #2: In this paper the authors investigate different properties of Topas copolymer, and of the same copolymer loaded with silica and with silica/hesperidin. The authors conclude that the addition of hesperidin improves the aging resistance of the material, since it may act as a radical scavenger. However, I think that their results do not support that conclusion. In addition, the presumed stabilizing effect is small.
- The authors obtain the hesperidin loaded materials by blending at 110º followed by vulcanization at 160º C. However, the melting temperature of hesperidin is 262º C. Hence, the flavonoid is not dissolved in the matrix, and should remain dispersed in the matrix retaining the same particle size distribution as in the feeding step. Considering the presence of only about 1% hesperidin, all of it in the form of crystallites, how can it be effective at all? I could understand some effect if hesperidin was dissolved in the matrix, but this doesn’t seem the case.
Response: We agree with the Reviewer that the melting temperature of hesperidin is equal to ~260 °C. It has been proved by the DSC analysis, for which the results are attached below. But in some scientific papers, also the flavonoids in the form of crystallites were able to slow down the aging process of polymeric materials. Regarding the amount of flavonoid used, as a rule, they are added in a small amount to the polymer matrix because of their price, but also of high efficiency. In these studies, we decided to use hesperidin in the amount of 1 phr, guided by available scientific articles, which also used the same or similar amount of antioxidants. The examples of literature on which we have operated are given below:
- Masek, A., Latos-Brozio, M., The Effect of Substances of Plant Origin on the Thermal and Thermo-Oxidative Ageing of Aliphatic Polyesters (PLA, PHA), Polymers, 2018, 10, 1252.
- Arrigo, R., Dintcheva, N.T., Natural Anti-oxidants for Bio-Polymeric Materials, Arch Chem Res., 2017, 1:2.
- Zaharescu, T., Zen, H.A., Marinescu, M., Scagliusi, S.R., Cardoso, E.C.L., Lugão, A.B., Prevention of degradation of γ-irradiated EPDM using phenolic antioxidants, Chemical Papers, 2016, 70(4).
- Samper, M.D., Fages, E., Fenollar, O., Boronat, T., Balart, R., The potential of flavonoids as natural antioxidants and UV light stabilizers for polypropylene, J. Appl. Polym. Sci., 2013, 129, 1707–1716.
- Latos-Brozio, M., Masek, A., Biodegradable Polyester Materials Containing Gallates, Polymers, 2020, 12, 677.
- Olejnik, O., Masek A., Kiersnowski, A., Thermal Analysis of Aliphatic Polyester Blends with Natural Antioxidants, Polymers, 2020, 12, 74.
Moreover, in order to check the form of hesperidin that was uploaded to the tested composites, we included the SEM analysis, which confirmed their crystalline structure.
- In the discussion of the surface free energy results, the authors observe very minor differences, that in my opinion could be simply attributed to experimental errors. They claim that hesperidin may enhance the amount of polar groups in the surface since it may migrate to the surface. However, the results obtained by ATR infrared spectroscopy shown in section 2.2 do not display any sign of the presence of hesperidin in the surface (C=O bands of hesperidin occur at about 1645 cm-1). In my opinion, the ATR results support that hesperidin has not been dispersed at the molecular level, it remains in crystalline form within the polymer matrix. Also, how can a substance that is not observable in the surface by ATR spectroscopy modify the surface energy of the samples?
Round 2
Reviewer 2 Report
As a minor mistake, in pg 8 line 236 the authors claim good water solubility for flavonoids, which is not true in my opnion since they cover a broad range of products with very different hydrophylicities (DOI: 10.1021/ie300211e)
Regarding the conclusions of the paper, I cannot believe that a product that is not detectable in the surface by FTIR, that according to the electron microscopy images (Figure 9) remains crystalline in spite of having been added in a very minor amount (about 1 wt%), can have a real effect on aging properties. But i prefer to pass by this work.
Author Response
Institute of Polymer and Dye Technology
Technical University of Lodz
90-924 Lodz, ul Stefanowskiego 12/16, Poland
Tel.: +48 42 631 32 23, Fax: +48 42 636 25 43
April 10, 2021
International Journal of Molecular Sciences
Dear Professor,
We are resubmitting our revised paper entitled “Influence of natural plant antioxidant on ageing process of Ethylene-norbornene Copolymer (Topas)” by with a request to reconsider it for publication in "International Journal of Molecular Sciences”.
We have carefully considered the second Reviewer comments. The manuscript was revised exactly according to these comments. The list of responses to the reviewer’s comments and corrections made in the manuscript are attached.
In the manuscript, the changes made based on the Reviewers' comments are marked in red.
The manuscript has not been previously published, is not currently submitted for review to any other journal, and will not be submitted elsewhere before a decision is made by this journal.
For correspondence please use the following information:
corresponding author: Anna Masek
Institute of Polymer and Dye Technology
Technical University of Lodz
90-924 Lodz, ul Stefanowskiego 12/16, Poland
Tel.: +48 42 631 32 93
Fax: +48 42 636 25 43
e-mail: anna.masek@p.lodz.pl
Yours sincerely,
PhD, Dsc Anna Masek
Answers to Reviewer #2 comments
Reviewer #2: As a minor mistake, in pg 8 line 236 the authors claim good water solubility for flavonoids, which is not true in my opinion since they cover a broad range of products with very different hydrophylicities (DOI: 10.1021/ie300211e).
Response: We agree with the Reviewer’s comment. We improved this part in our revised manuscript.
Reviewer #2: Regarding the conclusions of the paper, I cannot believe that a product that is not detectable in the surface by FTIR, that according to the electron microscopy images (Figure 9) remains crystalline in spite of having been added in a very minor amount (about 1 wt%), can have a real effect on aging properties. But i prefer to pass by this work.
Response: In our future research we are going to change the processing method and compare the effectiveness of this flavonoid in the form of crystallites a